# The Elevational Gradient of Bird Beta Diversity in the Meili Snow Mountains, Yunnan Province, China

**DOI:** 10.3390/ani13091567

**Published:** 2023-05-08

**Authors:** Shunyu Yao, Luming Liu, Pengfei Shan, Xiaojun Yang, Fei Wu

**Affiliations:** 1State Key Laboratory of Genetic Resources and Evolution &Yunnan Key Laboratory of Biodiversity and Ecological Conservation of Gaoligong Mountain, Kunming Institute of Zoology, Chinese Academy of Sciences, Kunming 650201, China; 2Kunming Natural History Museum of Zoology, Kunming Institute of Zoology, Chinese Academy of Sciences, Kunming 650223, China; 3College of Biological and Brewing Engineering, Taishan University, Tai’an 271000, China

**Keywords:** avian, beta diversity, turnover, limiting similarity

## Abstract

**Simple Summary:**

This study explores beta diversity patterns of birds, and their underlying processes with respect to taxonomic, phylogenetic, and functional facets, in the Meili Snow Mountains, Yunnan Province, China. A total of 3758 individuals representing 132 bird species were recorded during the fieldwork. We found distance–decay patterns on both the taxonomic and phylogenetic beta diversity along the elevational distance, with strong positive correlations with potential evapotranspiration and annual mean temperature. The turnover component dominated both the taxonomic and phylogenetic beta diversity. For both taxonomy and phylogeny, the limiting similarity dominated the turnover process in the Meili Snow Mountains. Our study aimed to provide information on understanding the mechanisms of beta diversity patterns in multiple dimensions.

**Abstract:**

Understanding the elevational patterns of beta diversity in mountain regions is a long-standing problem in biogeography and ecology. Previous research has generally focused on the taxonomy facet on a large scale, but was limited with regard to multi-facet beta diversity. Accordingly, we constructed a multi-dimensional (taxonomic/phylogenetic/functional) framework to analyze the underlying mechanisms of beta diversity. Within an approximately 2000 m altitudinal range (from 2027 m to 3944 m) along the eastern slope of the Meili Snow Mountains in Deqin County, Yunnan Province, China, we performed field surveys of breeding and non-breeding birds in September/2011 and May/2012, respectively. In total, 132 bird species were recorded during the fieldwork. The results indicated that taxonomic beta diversity contributed 56% of the bird species diversity, and its turnover process dominated the altitudinal pattern of taxon beta diversity; beta phylogenetic diversity contributed 42% of the bird phylogenetic diversity, and its turnover process also appeared to be stronger than the nestedness. For both taxonomy and phylogeny, the null models standardized measures (SES.β_sim_/SES.β_sne_/SES.β_sor_) of paired dissimilarities between elevation zones all showed statistically significant differences (*p* ≤ 0.05) and were higher than expected (SES.β > 0). However, standardized functional beta diversity showed convergence along the elevational gradient with no significant change. Moreover, the functional beta diversity contributed 50% of the bird functional diversity; there was no significant difference between the turnover and the nestedness-resultant component. Based on these results, we discerned that taxonomic and phylogenetic beta diversity patterns among the elevational zone were overdispersed, which indicated that limiting similarity dominated the turnover process among the bird species and phylogenetic communities in the Meili Snow Mountains.

## 1. Introduction

Understanding patterns of community assembly and their underlying processes remains a fundamental challenge in biogeography and ecology [1,2,3,4]. Traditionally, ecologists have examined variation in species diversity on local (alpha diversity) and regional (gamma diversity) scales [5]. However, variations in species diversity among sites (beta diversity) have gained increasing interest in the last three decades [6,7,8,9,10]. Beta diversity, originally defined by Whittaker (1960, 1972), is a measure of turnover in species diversity among sites and habitats or along gradients [11] and has also been widely used to illuminate the processes of diversity patterns (e.g., spatial, temporal) on multiple scales (e.g., ecological, evolutional) [7,12,13,14,15]. 

Beta diversity has been measured in many different ways in numerous studies [16], of which one of the most widely accepted approaches is Baselga’s approach [7], which is implemented by partitioning beta diversity into antithetical components: turnover and nestedness resultant [7,17]. This approach has been widely used for exploring elevational beta diversity. Generally, turnover occurs when there is replacement between distinct species at distinct sites; whereas the nestedness-resultant component occurs when biota of sites with lower numbers of species are subsets of biota with richer biota resulting from species loss and gain [7,18]. 

Revealing the determining processes that drive variation in community assembly and composition across sites is a fundamental and important challenge [2,10]. Theories and hypotheses on the underlying processes of the assembly, dynamics, and structure of ecological communities fall into two large groups, which are the niche-based (deterministic) hypothesis and the neutral theory. Starting with MacArthur [19], many other ecologists began to focus on the role of interspecific competition and environmental interactions in community assembly [20]. The niche-based hypothesis emphasizes the “ecological selection” process as the dominant mechanism, which leads to interspecific distinction or non-random patterns along environmental gradients. Conversely, the neutral theory emphasizes that a random dispersal process dominates the community structure [21]. Spatial variation (the distribution and abundance of species) among communities is an approach widely used to estimate the role that neutral or deterministic processes play as mechanisms underlying community assembly [22]. Exploring the underlying processes of beta diversity patterns can help to explain the roles of historical processes [23,24] and current environmental and anthropogenic activities [2,25].

Montane systems harbor a large and unique portion of the world’s biodiversity and are seen as biodiversity hotspots or as essential parts of other biodiversity hotspots [9] (e.g., Tropical Andes, Mountains of Central Asia, Mountains of Southwest China, etc.). Elevational gradients in montane systems provide an ideal experimental setting for exploring spatial patterns of biodiversity and community structures [26], and some of them contain great changes in topography, climate, and vegetation [2]. Thus, the exploration of variation in community assembly among sites along altitudinal gradients has gained increasing attention [9,27]. Historically, most beta-diversity studies have mainly focused on taxonomic variation in communities, which has been well studied and documented [16,28]. However, taxonomic beta diversity can only capture the extent of change among sites, and this is still a limit in uncovering evolutionarily developed or shared morphological similarities between species [3,29,30]. Whereas phylogenetic beta diversity can provide information on phylogenetic relatedness between samples of individual organisms inhabiting any two sites across space and time [30], phylogenetic measures explicitly integrate information on the evolutionary history of the species in a community and allow for the incorporation of information on trait conservatism [2,31]. Meanwhile, functional beta diversity enables comparisons between distinct geographic regions, which contain different faunas, by integrating functional traits with species data [32,33,34]. In recent years, the integration of phylogenetic and functional beta diversity with taxonomic beta diversity enabled researchers to evaluate the community structure in an evolutional and trait-based manner. It has also provided important insight into the processes and underlying mechanisms of community change, as well as the origin and maintenance of biodiversity [15,30,35,36]. 

In this study, we present bird data gathered from the eastern slope of the Meili Snow Mountains in Yunnan Province, China for the following purposes: (1) to identify the taxonomic, phylogenetic, and functional beta diversity patterns; as well as (2) the contribution of turnover and nestedness components to multi-facet beta diversity; (3) to conduct a comparison of multi-facet beta diversity with its turnover and nestedness components; (4) to examine the relative contributions of environmental factors and elevational distance to multi-facet beta diversity; and (5) to reveal the determining processes of patterns of beta diversity. In this study, we are aiming to test two hypotheses: 

**Hypothesis** **1:**
*There are no significant beta diversity patterns among bird community assemblies.*


**Hypothesis** **2:**
*The environmental factors and elevational distance have no significant correlations with patterns of beta diversity.*


## 2. Materials and Methods

### 2.1. Study Area

The study location was the eastern slope of the Meili Snow Mountains, which is located in Diqin County, Yunnan Province, China (98°43′~98°49′ E, 28°2′~28°32′ N). The Meili Snow Mountains are located between the Lancang River valley and the Nujiang River valley, forming part of the Hengduan Mountain. The mountain ridge forms the boundary between the Yunnan and Xizang Provinces [37]. The elevation range of the eastern slope of the Meili Snow Mountains is approximately 4500 m above sea level (the summit reaches 6435 m, and the valley is at approximately 1878 m). The relatively low latitude and great altitude range create a prominent vertical climatic gradient, so that the climate changes from a subtropical to a frigid climate [38]. Due to the high altitudes and complex climate, the vegetation in the Meili Snow Mountains also has a clear vertical distribution [39]. The vegetation changes with the elevation from a dry and warm valley shrub zone (below 2500 m) to warm coniferous forest (2500–3000 m). From 3000 to 3500 m, the vegetation is made up of coniferous and broad-leaf mixed forest; it is composed of cold coniferous forest and alpine shrub and meadow at altitudes from 3500 to 4500 m, and glacier and alpine talus above 4500 m [37]. The high altitudes, various types of vegetation, complex climate, and disparities in topography all reflect the high biodiversity in this area. 

### 2.2. Data Collection

#### 2.2.1. Field Sampling and Bird Surveys

We performed field surveys at three sample sites (Mingyong, Yubeng, Yongzhi) on the eastern slope of the Meili Snow Mountains with an approximate elevational range of 2000 m (from 2027 m to 3944 m) (Figure 1). We divided the elevational gradient into 8 bands in 250 m intervals (Appendix A). Bird communities were sampled using the point-count method, which is one of the most common methods for measuring species richness and relative abundance, especially for birds in complex forest ecosystems [40]. In this study, a total of 322 counts were obtained by two researchers in September, 2011 (the breeding season) and May, 2012 (the non-breeding season). Each survey point was visited twice during the sampling period, once in spring and once in winter. We conducted unlimited radius point counts in the peak period of bird activity between sunrise and 3.5 h after sunrise [9]. During this period, all birds seen and heard were recorded, and the horizontal distance from the central position of the survey point to each detected bird was estimated. All birds flying over the area and those with uncertain identification were removed from the analysis. The adjacent survey points were systematically located at least 200 m apart, and bird counting was only conducted on days with little or no wind, rain, or fog. We recorded the location of each point with a GPS receiver (Garmin-72H). The bird taxonomy used followed Yang et al. [41]. Nocturnal species, water birds, and raptors were excluded in the surveys. 

#### 2.2.2. Phylogenetic Data 

The phylogenetic data used in this study were obtained from the online database of global evolutionary distinctness in birds [42]. The original tree, Jetz’s tree, was assimilated from the genetic data of 6693 species of extant birds [42]. We chose the ‘Hackett backbone’: a set of 10,000 trees with 9993 operational taxonomic units to prune a subset tree of 132 birds of Meili using a pseudo-posterior distribution for subsequent phylogenetical calculation. The tree pruning measurement was performed using the “ape” [43], “ggtree” [44], and “geiger” [45] packages in R. 

#### 2.2.3. Functional Trait Data

Diets, foraging stratum, foraging time, body weight, and other functional attributes can largely reflect the species niche [42]. In this study, morphological and ecological traits were selected as the measurement indices of functional diversity. Morphological traits refer to body mass, while ecological traits include diets (invertebrates, vertebrates, fruits, nectar, seeds, other plants) and foraging stratum (ground, lower canopy, middle canopy, upper canopy, shrub layer). All the functional bird data were consecutive values downloaded from the online database of a global species-level compilation of key attributes, containing all extant bird species, derived from key literature sources (Appendix A) [42,46]. We then performed principal component analysis and retained the first two axes (89% of total inertia; Appendix A) for subsequent calculation. We also tested the phylogenetic signal for each trait using Pagel’s λ [47]. The significance of Pagel’s λ value was tested using the likelihood ratio test by comparing the log-likelihood values generated by λ [48]. λ ranges from 0 to 1. The closer λ is to 0, the longer the divergent time of the functional traits, and the greater the divergence of the functional trait compared to random Brownian motion; thus, the weaker the phylogenetic signal. On the contrary, the closer λ is to 1, the stronger the phylogenetic signal. All the functional traits in this study had significant phylogenetic signals (*p* < 0.001). The detection of three traits (diets, foraging stratum, and body mass) was conducted using the “phylosig” function of the “phytools” package in R [49]. 

#### 2.2.4. Environmental Data

We used five environmental variables to assess the elevational beta diversity of Meili: the annual mean temperature (Bio1), precipitation seasonality (Bio15), normalized difference vegetation index (NDVI), potential evapotranspiration (PET), and human footprint (HFP). Nineteen climatic predictors with a resolution of 30 arcseconds were obtained from Climatologies at High Resolution for the Earth’s Land Surface Areas [50]. We kept two variables (Bio1 and Bio15) based on Pearson’s correlation test to minimize collinearity. PET (30 arcseconds) was extracted from the Consortium for Spatial information [51]. NDVI (1 × 1 km) was extracted from the Resource and Environment Science and Data Center. Human footprint (HFP) (30 arcseconds) was obtained from The Last of Wild Project 2.0 [52]. The standardized environmental variables (mean value = 0, standard deviation = 1) were also transformed into a distance matrix using Euclidean distance for subsequent calculation.

### 2.3. Measuring Observed and Standardized Beta Diversity in Regard to Multi-Facet Beta Diversity

The incidence-based qualitative pairwise Sörensen dissimilarity index was used to measure the taxonomic, phylogenetic, and functional dissimilarities between montane communities [53]. The three dimensions’ pairwise dissimilarities were partitioned into turnover [54] and nestedness-resultant components. In this study, we used β_sor_, β_phylosor_, and β_funcsor_ to represent value of the overall beta diversity of taxonomic, phylogenetic, and functional beta diversity, measured as Sörensen dissimilarity [7,53]. We used β_sim_, β_phylosim_, and β_funcsim_ to represent the turnover component of taxonomic, phylogenetic, and functional beta diversity, measured as Simpson dissimilarity [7,17,54], and used β_sne_, β_phylosne_, and β_funcsne_ to represent the nestedness-resultant component of taxonomic, phylogenetic, and functional beta diversity, measured as nestedness-resultant fraction of Sörensen dissimilarity. The turnover and nestedness-resultant component of multiple dimensions were calculated based on the additive decomposing method [7]:$$\beta \mathrm{sor}=\frac{b+c}{2a+b+c}$$
$$\beta \mathrm{sim}=\frac{\min\left(b+c\right)}{a+\min\left(b+c\right)}$$
$$\beta \mathrm{sne}=\frac{\max\left(b,c\right)-\min\left(b,c\right)}{2a+\min\left(b,c\right)+\max\left(b,c\right)}\;\times \;\frac{a}{a+\min\left(b+c\right)}$$
where a is the number of species common to both sites, b is the number of species that occur at the first site but not at the second, and c is the number of species that occur at the second site but not at the first [7]. Functional dissimilarities were calculated using the “FD” package based on the convex-hull approach instead of the dendrogram-based approach [55,56]. Given the convex-hull approach’s methodological limitation, the observed and standardized trait beta-diversity measures were calculated for communities containing three or more species [57].

A null model for multi-facet beta diversity was constructed by randomly generating each set of community assembly data (keeping the same species richness in each assembly) 999 times. The dissimilarities in the standardized effect size (SES) of the taxonomic, phylogenetic, and functional dimension were calculated as follows:$$\mathrm{SES}=\frac{\mathrm{Observed}-\mathrm{Mean}\left(\mathrm{null}\right)}{\mathrm{SD}\left(\mathrm{null}\right)},$$
where Mean (null) and SD (null) are the mean value and standard deviation of each null distribution. The SES value was used to evaluate the degree of deviation between the observed dissimilarities and the expected values in random distributions and, furthermore, to assess the processes of community assemblies among montane zones. An SES value of more than 1.96 (*p* ≥ 0.975) indicates that the observed dissimilarities are higher than expected (overdispersal), and we found that competitive exclusion tended to dominate the community dissimilarity. An SES value of less than −1.96 (*p* ≤ 0.025) indicates that the observed dissimilarities are lower than expected (clustering) and environmental filtering tended to dominate the process through limiting dissimilarity. When the SES ranged from −1.96 to 1.96 (0.025 ≤ *p* ≤ 0.975), stochastic processes tended to dominate the dissimilarity [3]. The observed beta diversity of the three dimensions was calculated using the “beta. pair”, phylo. beta. pair”, and “functional. beta. pair” functions in the “betapart” package [58].

### 2.4. Statistical Analyses 

We used ANOSIM Test to test if there was a statistical difference between breeding bird communities and non-breeding bird communities. A two-sided Wilcoxon rank sum test was used to conduct the comparisons between the turnover and nestedness-resultant components and the total dissimilarities across the taxonomic, phylogenetic, and functional dimensions. We used Mantel’s test with 9999 permutations to examine the correlation between different components of beta diversity across the three dimensions and elevational as well as environmental distances using the “mantel.test” function in the “vegan” package [59]. The environmental distance matrix was generated using Euclidean distance with the “vegdist” function in the “vegan” package, and all the environmental variables were standardized (mean value = 0, standard deviation = 1) [60]. We then used hierarchical partitioning to examine the average independent effect of each variable contributing to multi-facet dissimilarities between community assemblies with the “hier.part” package [61]. We also used linear regression to examine the relationship between elevational distance and the observed multi-facet and standardized dissimilarities. All the statistical analyses were performed using R Statistical Software (v4.1.2) [62].

## 3. Results

We recorded 3758 individuals representing 132 bird species during our fieldwork (Appendix A). According to an ANOSIM Test, there was no significant difference between breeding bird communities and non-breeding bird communities (*p* = 0.83). Thus, we combined the bird communities of two season for subsequent calculation. According to the beta diversity across multiple dimensions, the turnover component dominated both the taxonomic and phylogenetic beta diversity, while there was no significant difference between the turnover and the nestedness-resultant component of functional beta diversity (Table 1).

Table 2 shows the results of turnover, nestedness-resultant components, and total dissimilarities of taxonomic, phylogenetic, and functional beta diversity in the Meili Snow Mountains.

### 3.1. Comparison between Turnover, Nestedness-Resultant Components, and Total Dissimilarities across Multiple Dimensions

According to the results of the two-sided Wilcoxon test (Table 2, Figure 2), the turnover component dominated the altitudinal pattern of beta diversity in both the taxonomic and phylogenetic dimensions. On the contrary, trait-based dissimilarities did not show a significant pattern along the elevational gradient, and the turnover component tended to have the same effect as the nestedness-resultant component. Our comparisons between the diversity dimensions showed that the total dissimilarities in the taxonomic dimension were significantly higher than those of the phylogenetic and functional facets, while the total dissimilarities between the phylogenetic and functional dimensions were also significant. Similarly, the taxonomic turnover component was the highest among the three dimensions, and the turnover between the phylogenetic and functional dimensions was also significant. There was no significant relationship between the nestedness-resultant dissimilarities across the three dimensions.

### 3.2. The Variation in Observed Beta Diversity along Elevational Distance

According to the results of Mantel’s test, the elevational distance was closely correlated with the total dissimilarities (β_sor_, β_phylosor_) and turnover component (β_sim_, β_phylosim_) for the taxonomic and phylogenetic facets, and both the taxonomic and phylogenetic facets showed a consistent montane linear pattern. On the contrary, elevational distance had no significant linear relationship with the nestedness-resultant component (β_sne_, β_phylosne_) or functional beta diversity (Table 3, Figure 3).

### 3.3. Environmental Factors

According to Mantel’s test, strong correlation was found only between the environmental parameters and total dissimilarities (β_sor_, β_phylosor_) on the one hand, and turnover components (β_sim_, β_phylosim_) on the other hand, across the taxonomic and phylogenetic dimensions, as was the case of the elevational distance. On the contrary, there was no significant relationship between the environmental distance matrix and functional beta diversity (Table 4). Therefore, we used hierarchical partitioning focusing only on the taxonomic and phylogenetic facets to identify the average independent effect of each variable contributing to the processes of community assembly. The results of hierarchical partitioning indicated that the best environmental predictors were distinct, varying between different components (β_sim_ & β_sne_; β_phylosim_ & β_phylosne_) but had consistency in the taxonomic and phylogenetic dimensions and also in their total dissimilarities. Among the five environmental variables, potential evapotranspiration (PET) and annual mean temperature (Bio1) contributed the most to the elevational patterns of both taxonomic and phylogenetic total dissimilarity and the turnover component (Figure 4).

### 3.4. Standardized Pairwise Beta Diversity among Elevation Gradient

According to the results of the linear regressions for the taxonomic and phylogenetic facets, the standardized total dissimilarities (β_sor_, β_phylosor_) and turnover component (β_sim_, β_phylosim_) showed consistently positive significance (*p* < 0.01) as the elevational distance increased, while the nestedness-resultant components had no significant correlation with the elevation distance. For the functional facet, no consistently significant linear pattern was found with the variation in elevation: 90% of the standardized functional turnover was less than −1.96, thereby being lower than expected; 39% of the standardized functional nestedness-resultant component was higher than 1.96, thereby being higher than expected; and 71% of the total functional dissimilarities were less than −1.96, thereby being lower than expected (Figure 5).

## 4. Discussion

In this study, the pattern of functional beta diversity, showing no significant pattern between bird community assemblies, supported our Hypothesis 1. However, the patterns of taxonomic and phylogenetic beta diversity were inconsistent with our Hypothesis 1, as they both showed a distance–decay pattern along the elevational zones. Similarly, both environmental and elevational distance had no significant correlation with functional beta diversity, which supported our Hypothesis 2. However, inconsistent with Hypothesis 2, both elevational and environmental distance had strong positive correlations with taxonomic and phylogenetic beta diversity. Turnover and nestedness-resultant processes were viewed as two antithetic processes shaping community structure [7]. Taxonomic beta diversity is generally seen as a species’ reaction to its current environment [3]. According to our results, the comparison between taxonomic components revealed that spatial turnover strongly dominated the taxonomic beta diversity, indicating frequent species replacement among bird assemblies rather than species gain/loss that shaped the bird communities’ structure along the elevational gradient [7]. This pattern is consistent with many other taxa and regions [7,12,15]. For example, spatial turnover was found to be a relatively minor component in the case of plants, invertebrates, and vertebrates along the climatic gradient in Britain [12], and similar conditions were also found in the taxonomic beta diversity of passerines, rodents, and ants along eastern slope of Mt. Segrila [15], as well as birds, plants, and butterfly assemblies in Switzerland due to variations in the environment (a steep climatic gradient and a change in vegetation) [1]. Phylogenetic beta diversity quantifies how phylogenetic relationships between species change across space [30]. Similar to the taxonomic beta diversity, the phylogenetic dissimilarities between bird assemblies increased as the elevation distance increased, and the turnover component dominated the phylogenetic beta diversity. A similar result was found in the case of wetland birds in North China [63]. Nevertheless, this result was inconsistent with that of bird assemblies on Mt. Segrila and amphibian communities on Mt. Emei, where the nestedness-resultant component dominated the phylogenetic beta diversity [15]. The uplift of the Qinghai–Tibet plateau led to the speciation of a series of taxa at a high elevation, which formed specific Qinghai–Tibet plateau fauna [64,65]. As the nestedness-resultant component occurs when biota of sites with lower numbers of species are subsets of biota with richer biota resulting from species loss and gain, the specific bird fauna at high elevations on Mt. Segrila have formed a species pool which was an origin of species for lower elevations. Similarly, amphibian assemblies in low lands also have formed a species pool which was an origin of species for higher elevations. Here in our study, we suggest that phylogenetic dissimilarities may be influenced by both high- and low-elevation taxa, thus leading to a turnover-dominated pattern (e.g., low elevation species: scientific name: *Dicaeum ignipectus*, common name: fire-breasted flowerpecker; scientific name: *Cephalopyrus flammiceps,* common name: fire-capped tit; and high elevation species: scientific name: *Ithaginis cruentus*, common name: blood pheasant; scientific name: *Trochalopteron elliotii*, common name: Elliot’s laughingthrush). Functional beta diversity indicates the extent of adaption of a species to variation in habitats and environments [55]. Inconsistent with the pattern of taxon and phylogeny, our comparison results showed that there was no significant difference between the turnover and nestedness components. As the elevational range in our study was not large, the vegetation of the elevational zones constituted a mixture of similar forest types, and the adaption of bird assemblies among the communities to the change in habitats was relatively stable.

In comparisons between the taxonomic, phylogenetic, and functional facets, high taxonomic, high phylogenetic, and low functional relationships of turnover (β_sim_, β_phylosim_, β_funcsim_) and total dissimilarities (β_sor_, β_phylosor_, β_funcsor_) were found for the Meili Snow Mountains. In a study of hummingbirds in South America [3], Weinstein et al. examined a set of hypotheses encompassing combinations of different extents of multi-facet beta diversity in accordance with different environmental conditions. The high taxonomic, high phylogenetic, and low functional beta diversity observed in our study confirmed their hypothesis, that different elevational zones have similar forest types, providing a relatively stable adaptation environment for bird assemblies and indicating that species from different communities and different clades have trait similarities. Therefore, although there was frequent species replacement among the bird assemblies, and despite the fact that they had different evolutional histories, the functional traits of these bird assemblies were still convergent [55].

The similarity between two species assemblies tends to be stronger when they are closer in spatial distance and weaker when they are further from one another, which is defined as a distance–decay pattern [13,66,67,68]. We found a distance–decay pattern in both the taxonomic and phylogenetic beta dissimilarities observed in the Meili Snow Mountains, while no significant pattern was observed in the observed functional beta dissimilarities. According to Mantel’s test, both elevation and environmental distance have a strong correlation with taxonomic and phylogenetic beta diversity, whereas functional beta diversity has a weak response to either elevation or environmental distance, indicating that there are non-random interactions between variations in environment and bird assemblies. Similar patterns were also found in other taxa and regions [13,68,69]. Potential evapotranspiration (PET) and annual mean temperature (Bio1) contributed the most to the total dissimilarities and turnover component of the taxonomic and phylogenetic facets, and NDVI explained most of the nestedness-resultant component. According to the elevational patterns of environmental variables, both potential evapotranspiration (PET) and annual mean temperature (Bio1) decreased along the elevational gradient, contributing to the increase in the environmental selection effect (less food supply and a tougher habitat) and then leading to specific fauna and frequent species replacement. However, environmental distance had no significant correlation with functional beta diversity. The similar mixture of forest types among elevational zones may be the reason for the weak response of functional beta diversity to the environmental variables; thus, the best predictors of dissimilarities in regard to functional dissimilarity still need to be tested.

We examined the pattern of standardized multi-dimensional beta diversity via linear regression to assess the underlying process of community structuring throughout the elevational zones. According to our results, for both the taxonomic and phylogenetic facets, the standardized turnover showed a significant increase along the elevational zones, implying that the effect of limiting similarity was gradually enhanced with the increased elevation, while the effect of environmental filtering decreased. The SES value (taxonomic and phylogenetic dimensions) of less than −1.96 at a lower elevation indicated a pattern of convergence contributing to abiotic filters (shared tolerance between coexisting species is high), while the SES value larger than 1.96 at a higher elevation indicated that negative interspecies competition gradually increased over a larger elevational distance. There was also a random distribution of standardized beta diversity in both taxonomic and phylogenetic dimensions over a medium elevational distance, suggesting that a dynamic balancing effect may occur between environmental filtering and limiting similarity. The nestedness-resultant component (β_sne_, β_phylosne_) showed an opposite pattern to turnover but with minor variation along the elevation gradient. However, the result for standardized functional beta diversity showed a significant convergent pattern along the elevational gradient. Ninety percent of the functional standardized turnover and seventy percent of the standardized total dissimilarity values were less than −1.96 in the lower, middle, and larger elevational zones, implying that in a frequent replacement process, high- and low-elevation species both have similar traits, or the ecological niche in each elevation zone is nearly filled up [70]; therefore, the speed of trait diversification decreases. However, we cannot be sure of the exact process contributing to the functional elevation pattern, and it may be that a joint effect contributes to several mechanisms. In closing, we obtained some interesting results, but there are still some short-comings which should be addressed. First, although we performed intensive and standardized field surveys, the count number of each elevational band was not comparable. Second, although our study contained two breeding seasons, we only did a single-year survey on a spatial scale. Thus, the dissimilarities among bird community assemblies on a temporal scale can be further explored.

## 5. Conclusions

We performed a series of analyses in this study to understand the elevational beta diversity pattern of the Meili Snow Mountains. By partitioning beta diversity into multiple dimensions (taxonomic, phylogenetic, and trait), comparing patterns within and between these dimensions along the elevational gradient, as well as revealing the underlying processes, we found the following: (1) The turnover component dominated both taxonomic and phylogenetic beta diversity along the elevation gradient. However, for functional beta diversity, there was no significant difference between the turnover and nestedness-resultant processes. (2) Comparisons between the three dimensions showed that total dissimilarities and turnover process of the taxon appeared to be the highest contributors, and the deviation showed strong significance; conversely, despite the fact that the deviation was weak, the nestedness-resultant component of the trait dimension was higher than that of the taxon and phylogeny. (3) Elevational distance and environmental variables had the same variation in terms of the taxonomic and phylogenetic total dissimilarities and turnover component but had no significant correlation with functional beta diversity. Among the five environmental variables, potential evapotranspiration (PET) and annual mean temperature (Bio1) contributed the most towards explaining the elevational patterns of taxonomic and phylogenetic beta diversity, while NDVI explained most of the nestedness component. (4) For the taxonomic and phylogenetic elevational patterns, the effect of limiting similarity gradually increased with increased elevation, while the process underlying the functional beta diversity pattern still remains unclear.

## Figures and Tables

**Figure 1 animals-13-01567-f001:**
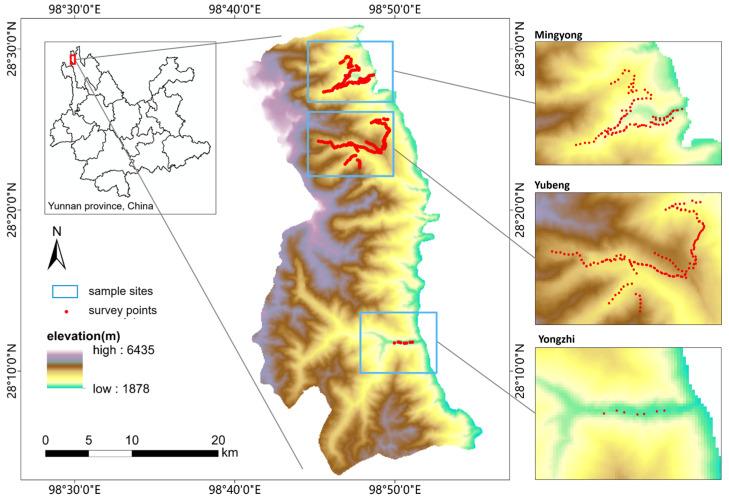
Study area, sample sites (blue frame) and survey counts (red solid circles) of our field survey of the eastern slope of the Meili Snow Mountains, Yunnan, China.

**Figure 2 animals-13-01567-f002:**
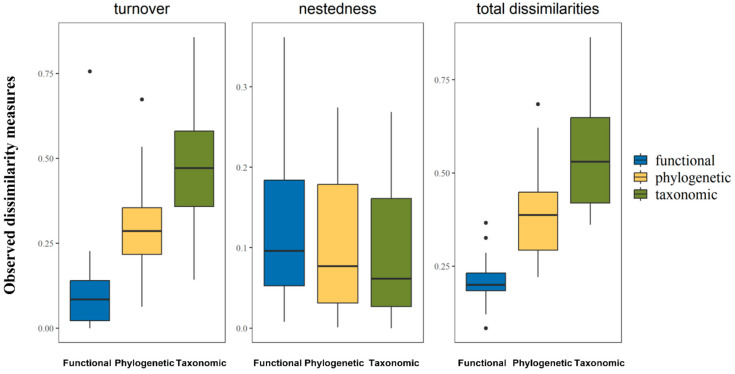
Results of the two-sided Wilcoxon rank sum test for a comparison between total dissimilarities, and turnover and nestedness components across taxonomic (green), phylogenetic (yellow), and functional (blue) dimensions.

**Figure 3 animals-13-01567-f003:**
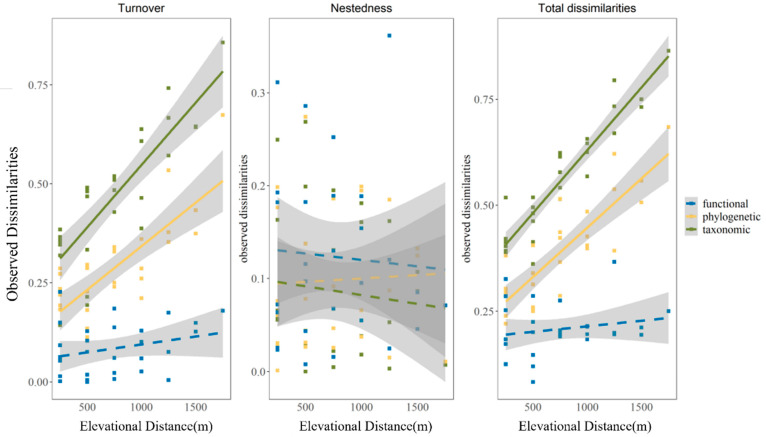
Linear regression between elevation and observed dissimilarities (turnover, nestedness-resultant component, and total dissimilarities) across taxonomic (green), phylogenetic (yellow) and functional (blue)dimensions. Bold lines in all colors indicate a significant linear pattern (*p* < 0.01); dashed lines in all colors imply that there are no significant patterns. Details are in Appendix A.

**Figure 4 animals-13-01567-f004:**
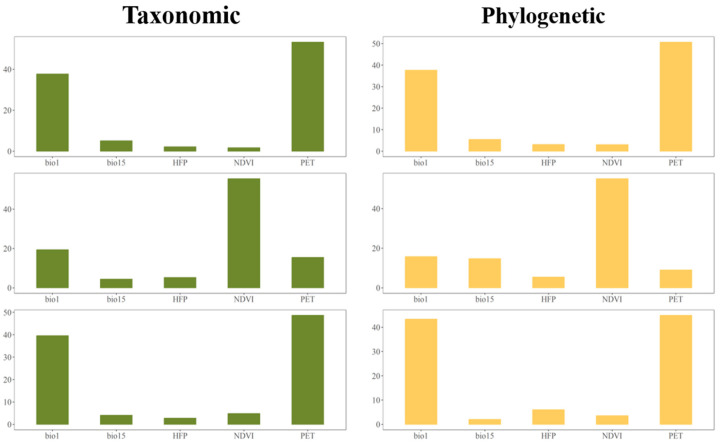
The independent contribution of each environmental predictor to different component of taxonomic (green) and phylogenetic (yellow) beta diversity. Bio1: annual mean temperature; Bio15: precipitation seasonality; HFP: human footprints; NDVI: normalized difference vegetation index; PET: potential evapotranspiration.

**Figure 5 animals-13-01567-f005:**
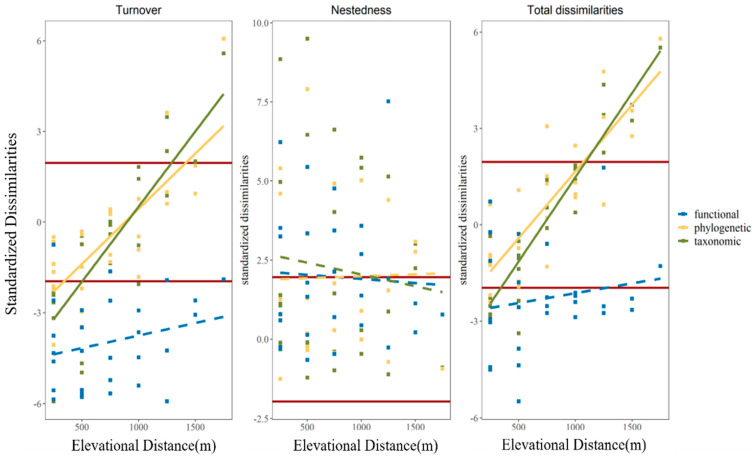
Linear regression between elevation and standardized dissimilarities (turnover, nestedness-resultant component, and total dissimilarities) across taxonomic (green), phylogenetic (yellow), and functional (blue) dimensions. Bold lines in all colors indicate a significant linear pattern (*p* < 0.01); dashed lines in all colors imply that there is no significant pattern. The zones between two horizontal red lines in each plot represent a range from −1.96 to 1.96. Details are in Appendix A.

**Table 1 animals-13-01567-t001:** The results of turnover, nestedness-resultant components, and total dissimilarities of taxonomic, phylogenetic, and functional beta diversity in the Meili Snow Mountains.

	Taxonomic	Phylogenetic	Functional
Turnover	0.68	0.54	0.25
Nestedness-resultant	0.06	0.1	0.24
Total dissimilarities	0.74	0.64	0.48

**Table 2 animals-13-01567-t002:** The results of the two-sided Wilcoxon rank sum test for the comparison between total dissimilarities (β_sor_, β_phylosor_, β_funcsor_), turnover (β_sim_, β_phylosim_, β_funcsim_), and nestedness-resultant component (β_sne_, β_phylosne_, β_funcsne_) across taxonomic, phylogenetic, and functional dimensions. *p* values in bold indicate strongly significant correlations, “/” means there is no significant relationship.

	|Z-Score|	*p*-Values	Patterns
β_sim_ vs. β_sne_	7.36	**<0.001**	β_sim_ > β_sne_
β_phylosim_ vs. β_phylosne_	5.4	**<0.001**	β_phylosim_ > β_phylosne_
β_funcsim_ vs. β_funcsne_	0.24	0.21	/
β_sim_ vs. β_phylosim_	4.05	**<0.001**	β_sim_ > β_phylosim_
β_sim_ vs. β_funcsim_	5.84	**<0.001**	β_sim_ > β_funcsim_
β_sne_ vs. β_phylosne_	0.71	0.48	/
β_sne_ vs. β_funcsne_	1.42	0.17	/
β_phylosim_ vs. β_funcsim_	5.88	**<0.001**	β_phylosim_ > β_funcsim_
β_phylosne_ vs. β_funcsne_	0.77	0.44	/
β_sor_ vs. β_phylosor_	4.04	**<0.001**	β_sor_ > β_phylosor_
β_sor_ vs. β_funcsor_	6.4	**<0.001**	β_sor_ > β_funcsor_
β_phylosor_ vs. β_funcsor_	5.66	**<0.001**	β_phylosor_ > β_funcsor_

**Table 3 animals-13-01567-t003:** Mantel’s test with 9999 permutations between elevational distance and total dissimilarity, turnover, and nestedness-resultant component across taxonomic, phylogenetic, and functional dimensions. The r value in bold indicates a strong correlation (*p* < 0.01) between elevation distance and beta diversity components.

Facet	Taxonomic	Phylogenetic	Functional
Turnover	**r = 0.88**	**r = 0.83**	r = 0.30
Nestedness	r = −0.19	r = −0.02	r = −0.05
Total dissimilarities	**r = 0.90**	**r = 0.85**	r = 0.19

**Table 4 animals-13-01567-t004:** Mantel’s test with 9999 permutations for a comparison between environmental distance and total dissimilarity, turnover, and nestedness-resultant component across taxonomic, phylogenetic, and functional dimensions. The r value in bold indicate a strong correlation (*p* < 0.01) between elevation distance and beta diversity components.

Facet	Taxonomic	Phylogenetic	Functional
Turnover	**r = 0.80**	**r = 0.73**	r = 0.31
Nestedness	r = 0.19	r = 0.02	r = 0.047
Total dissimilarities	**r = 0.87**	**r = 0.845**	r = 0.194

## Data Availability

The data presented in this study are available in the Appendix A.

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
