# Peer review of "The Elevational Gradient of Bird Beta Diversity in the Meili Snow Mountains, Yunnan Province, China"

_animals, 2023, doi:10.3390/ani13091567_

Round 1
Reviewer 1 Report
This is a very interesting manuscript, which focuses on a fundamental and fascinating topic in ecology - the drivers of bird beta diversity across an altitudinal gradient. I believe that the methodology is adequate, although there were times when I felt that there was a lack of information about certain procedures.
As a non-native English speaker, I usually only require that the text is fully understandable. However, in this case, I found that the manuscript was very difficult to follow at times. I have provided some comments and suggestions in the attached PDF, but I would like to emphasize that a thorough language review is necessary. Additionally, please ensure that the figures are adjusted, and the references are checked.

Reviewer 2 Report
This paper analyses a rich data set on bird species across an elevational gradient. The point counts were conducted in summer and the fall but there is no description of how the data from the two seasons were used. Were they just combined? Because of altitudinal migration present in many montane systems and then the possible influx of latitudinal migrants especially at lower elevations it should be clear which avian community they are analysing. It would also help the reader to provide the range of environmental variables that occur within the study area to contextualize this work (e.g., is it tropical, temperate?). I think the analyses are competent and the results interesting and probably valid. There are many issues with verb tenses that make for a confusing presentation along the way. I have provided some partial guidance to clarify some of these sentences.
L. 13-14. I don’t understand this sentence.
L. 29. Instead of ‘showed overdispersed’ please use ‘were overdispersed’
L. 39. ‘increasing attention’
L. 45. Use ‘limited’ instead of ‘limit’. Also in this sentence ‘processes’ instead of ‘process’. And it is hard to know what is meant by ‘shared morphological similarities’..Do the authors mean how important shared morphology is to understanding taxonomic patterns in beta diversity?
L. 49. Use ‘process’
L. 50. Use ‘traits’
L. 55. Delete ‘in amounts of studies’ . Also replace ‘by’ with ‘in’ (as in ‘in many different ways’)
L. 62. Use ‘reveal’
L. 74. Give altitudinal extent of the gradient here.
L. 81. Give the environmental variables that you are using here.
L. 92. I am confused about why there is a mention of amphibians when I thought the paper was about birds.
L. 109. Use ‘conducting point counts’ rather than ‘staying at point counts’
L. 112. Need ‘were recorded’ at end of the sentence that states the variables that you measured. Also in this sentence use plural of photo ‘photos’
L. 130. ‘kept’ instead of ‘keep’
Results. First paragraph. Please break into separate sentences for each component of diversity and use parallel construction giving the beta and then the percentage for each component. Right now it is confusing to read.
L. 234. ‘results of the Mantel test’
L. 236. Delete ‘they’ if that is what was intended..that both the taxonomic and phylogenetic facets showed consistent montane linear patterns…
L. 258. No need to use acronyms as this creates additional confusion. Use full names of environmental variables here.
L. 324. Use ‘hummingbirds’
L. 331. Use ‘different evolutionary histories’
L. 376. ‘analyses’
Round 2
Reviewer 1 Report
I would like to commend the authors on the substantial improvements made to the manuscript titled "The elevational gradient of bird beta diversity on Meili Snow Mountain, Yunnan Province, China." While there are still some typos and sentences that require clarification (please refer to the attached PDF), the document now reads more smoothly and is more enjoyable to read.
The introduction is satisfactory, but I would suggest that in the final paragraph where the authors outline their objectives and expectations, they should consider formulating their ideas as hypotheses. This would be beneficial because, during the discussion, several hypotheses are mentioned that were not explicitly presented in the manuscript.
In relation to the methodology, the authors seem to have adequately addressed the objectives of the study. However, it would be helpful if the authors could provide further clarification on how they accounted for the lack of independence and potential pseudo-replication of the data. Specifically, since the survey was conducted using multiple focal points with unlimited radius, and the focal points were only 200m apart, there may be significant overlap in the surveyed area between points. It would be useful for the authors to explain how they addressed this issue in their analysis.
The authors have done a commendable job presenting the results in a clear and concise manner, and the figures and tables are appropriately used to support the findings. The discussion section provides a comprehensive interpretation of the results, but would benefit from a more thorough revision of the text to improve its clarity and flow.
I would like to express my appreciation to the authors for considering my comments and suggestions to improve their manuscript. I understand that writing in English can be challenging, particularly for non-native speakers. I hope my feedback was helpful and contributed to enhancing the quality of their work.
Good luck!

Author Response
Please see the attachment. Thank you very much for your help.
